# Automotive Object Detection under Adversarial Attacks: A Robust Two-Stage Training Framework

## Abstract

Deep learning-based object detectors are widely used in autonomous driving, but remain vulnerable to adversarial attacks. In this paper, we propose a lightweight and modular two-stage training framework that enhances adversarial robustness by integrating a binary adversarial classifier into the backbone of a Faster Region-Based Convolutional Neural Network (Faster R-CNN). The classifier is trained in two phases: (1) freezing the detector while learning to distinguish adversarial inputs, and (2) joint fine-tuning to refine performance. Our method achieves over 90% accuracy in detecting adversarial examples and generalizes well across white-box attack types and strengths, including the Fast Gradient Sign Method (FGSM), Basic Iterative Method (BIM), and Projected Gradient Descent (PGD). The model demonstrated strong zero-shot transferability and achieved further gains after fine-tuning on the target datasets. We evaluated the approach on the Berkeley DeepDrive 100K (BDD100K) dataset and further demonstrated strong cross-dataset robustness on KITTI and nuScenes: the model transfers effectively in a zero-shot setting and achieves AUROC above 0.98 under BIM and PGD attacks after fine-tuning. The proposed defense improves adversarial perturbation detection with minimal impact on object detection accuracy, making it suitable for real-world deployment in automotive systems.

## 1 Introduction

Object detection plays a critical role in many modern applications such as autonomous driving, surveillance, and robotics. Despite significant advances achieved by deep learning-based detectors, they remain vulnerable to adversarial perturbations: small, often imperceptible modifications to input images that can cause detectors to miss objects or output incorrect bounding boxes. These vulnerabilities are especially concerning in safety-critical scenarios, where detection failures can lead to severe consequences. Figure 1 illustrates this problem, showing how our model detects such attacks and provides a confidence score.

We focus on digital adversarial perturbations applied directly to input images at test time. While physical-world attacks (e.g., adversarial patches or 3D modifications) have been explored, digital attacks remain a practical and immediate threat in real-world pipelines, particularly in autonomous systems where image data may be processed or transmitted through intermediate modules, such as in vehicle-to-infrastructure (V2I) or cloud-based architectures. These attacks can originate from compromised sensors, corrupted update mechanisms, or manipulated simulation data used in training. Beyond technical risks, adversarial examples challenge the credibility of autonomous vision systems, influencing public trust, regulation, and commercial adoption.

Studying digital attacks is a necessary step in understanding and mitigating vulnerabilities in object detection. Importantly, we assume a white-box or transfer-based adversary with access to the object detector but not to the proposed detection module. Physical-world perturbations and fully adaptive attackers are considered out of the scope of this study.

Although adversarial defenses for image classification are well studied, object detection poses unique challenges due to its multi-task nature (classification + localization) and spatial sensitivity. Most existing defenses involve retraining the detector, applying input transformations, or modifying

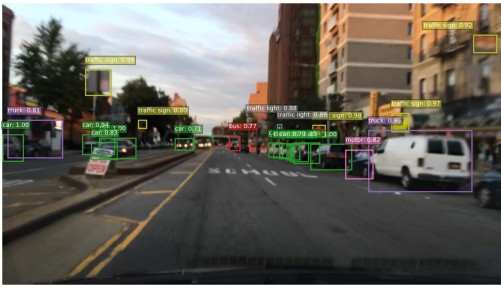

(a) Clean input — no attack detected (score: 0.22)     (b) Adversarial input — attack detected (score: 0.75)

Figure 1: Detection results on clean (top) and adversarial (bottom) inputs from the BDD100K test set. In the clean image, vehicles and traffic elements are correctly detected with high confidence. The adversarial image appears visually identical but causes misclassifications (e.g., cars detected as trucks or buses) and spurious detections (e.g., traffic signs in trees). Our adversarial classifier (highlighted in red in Figure 2) flags the input as adversarial despite high detector confidence, demonstrating the effectiveness of the proposed defense.

the architecture, approaches that are often attack-specific, computationally heavy, or incompatible with pre-trained models.

**In this work, we propose a lightweight attack-agnostic defense that augments existing detectors without altering them.** Specifically, we introduce a modular binary classifier that operates on intermediate features of a Faster R-CNN backbone. Through a two-phase training strategy, first training the classifier with a frozen detector, then jointly fine-tuning, we enable early detection of adversarial inputs with minimal computational cost. Our architecture is illustrated in Figure 2, showing how the adversarial classifier integrates into the detection pipeline. The design preserves clean detection accuracy while providing robustness against common white-box attacks (FGSM, BIM, PGD) and generalizes across datasets. We extensively evaluated our method on BDD100K and further validated its adaptability on KITTI and nuScenes, showing both strong zero-shot transferability and effective domain adaptation.

The contributions of this work are as follows:

- A modular adversarial detector that leverages intermediate features of Faster R-CNN without modifying the detection architecture.

- A two-phase training procedure that balances adversarial robustness with clean input performance.

- Extensive evaluation across multiple types and strengths of attacks, showing lightweight and attack-agnostic robustness.

- Cross-dataset validation on BDD100K, KITTI, and nuScenes, demonstrating both strong zero-shot generalization and effective adaptation through fine-tuning.

The remainder of this paper is organized as follows. Section 2 reviews prior work on adversarial attacks and defenses in object detection. Section 3 details our classifier design and training procedure. Section 4 presents experimental results and analysis. Section 6 discusses implications and limitations, followed by conclusions in Section 7.

## 2 RELATED WORK

**Adversarial Attacks and Defenses in Computer Vision.** Adversarial attacks, first shown to mislead image classifiers Szegedy et al. (2013); Goodfellow et al. (2014), have been extended to object detection where they cause dangerous failures like missed detections or shifted bounding boxes Xie et al. (2017); Chen et al. (2019). Defenses against these attacks often employ adversarial training Madry et al. (2017), which, while effective, is computationally expensive and can degrade perfor-

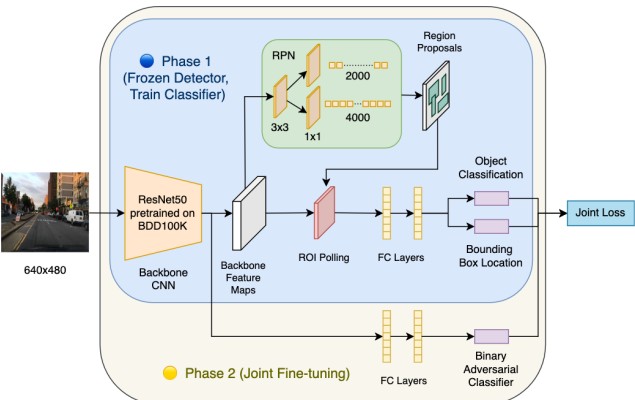

Figure 2: Overview of the proposed architecture. A binary adversarial classifier is attached to intermediate features of the Faster R-CNN backbone. In Phase 1, the detector is frozen (blue box) and only the classifier is trained. In Phase 2, all components are fine-tuned jointly. A shared loss combines detection and adversarial classification objectives.

mance on clean inputs. Other approaches include architectural modifications or input transformations Liu et al. (2019); Song et al. (2018), which often lack generality.

**Detection-Based Defenses.** An alternative strategy is to detect adversarial inputs using auxiliary classifiers or statistical tests Metzen et al. (2017); Feinman et al. (2017); Xu et al. (2017). While a common approach in image classification, this is more challenging for object detection due to its complex, multitask nature Guo et al. (2020). Our approach builds on this by integrating a lightweight, modular binary classifier into the detector's backbone.

**Adversarial Robustness in Automotive Applications.** In autonomous driving, adversarial robustness is critical but underexplored. Early work focused on physical attacks, such as perturbed traffic signs Eykholt et al. (2018). However, digital attacks pose a significant threat in systems with remote computation or V2X communication Zhang et al. (2021). While large-scale datasets like BDD100K Yu et al. (2020), KITTI Geiger et al. (2012), and nuScenes Caesar et al. (2020) provide diverse benchmarks, defenses for full-scene object detection in these contexts are scarce. Our work addresses this gap by targeting a generalizable defense that is integrated directly into the perception backbone, unlike prior work that focuses on specific attacks or datasets.

**Our Contribution.** We propose a lightweight, modular binary classifier integrated into a Faster R-CNN detector Xu et al. (2020). Our method is attack-agnostic and maintains the detector's original architecture. We introduce a two-phase training strategy that improves adversarial detection accuracy without the computational cost of full adversarial retraining or impacting performance on clean data.

## 3 METHODOLOGY

Our method enhances adversarial robustness in object detection by attaching a lightweight binary classifier to a pre-trained Faster R-CNN, enabling detection of adversarial inputs without modifying the detector architecture.

**Architecture.** We build on the standard Faster R-CNN, which consists of a *backbone* for feature extraction, a *Region Proposal Network (RPN)* for candidate regions, and task-specific *heads* for classification and bounding box regression. As shown in Figure 2, the binary classifier is connected to the final convolutional features of the backbone. Its role is to classify each input as clean or adversarial while leaving the detection pipeline unchanged.

**Two-Phase Training.** To stabilize training and balance robustness with clean accuracy, we adopt a two-phase strategy:

1. **Phase 1 (Frozen Detector).** The backbone, RPN, and detection heads remain fixed. Only the adversarial classifier is trained on balanced clean/adversarial examples. This ensures convergence without disrupting pre-trained detection features.

2. **Phase 2 (Joint Fine-tuning).** The detector is unfrozen and optimized jointly with the adversarial classifier, allowing shared features to adapt for more challenging perturbations while maintaining detection accuracy.

**Loss Function.** Optimization is performed using a joint objective:

$$\mathcal{L}_{\text{total}} = \mathcal{L}_{\text{det}} + \lambda_{\text{aux}} \cdot \mathcal{L}_{\text{adv}}, \tag{1}$$

where $\mathcal{L}_{\text{det}}$ is the standard Faster R-CNN detection loss, $\mathcal{L}_{\text{adv}}$ is the binary cross-entropy loss for adversarial classification, and $\lambda_{\text{aux}}$ balances the two tasks.

**Adversarial Training Samples.** Adversarial inputs are generated using FGSM with varying magnitudes, ensuring generalization across attack strengths. The classifier's decision threshold is calibrated post-training to balance false positives and false negatives for reliable deployment.

This design keeps the adversarial classifier lightweight and detector-agnostic, enabling seamless integration into object detection pipelines with minimal overhead.

## 4 EXPERIMENTAL SETUP

### 4.1 DATASET AND MODEL CONFIGURATION

We evaluated our framework primarily on the BDD100K dataset, which is a large-scale benchmark for autonomous driving with diverse scenes and weather conditions. To reduce computational cost, we randomly sampled approximately 2% of the training data (1,397 images) and applied a 70/20/10 train–validation–test split. We also included experiments on KITTI and nuScenes for cross-dataset robustness, using similarly sized subsets (below 1,500 images) and the same split ratios to ensure consistent data scale across benchmarks. All images were resized from $1280 \times 720$ to $640 \times 480$ to match the input resolution of the COCO-pretrained Faster R-CNN backbone. We restricted evaluation to 10 key object categories relevant to driving.

The base detector is a Faster R-CNN with a ResNet-50 backbone and FPN, pre-trained on COCO. We adapted the model using a staged transfer learning schedule of 10, 10, and 20 epochs to stabilize training on limited data. The auxiliary adversarial classification head applies global average pooling to the final backbone features, followed by a fully connected layer with 256 units and ReLU, a dropout layer (rate 0.3), and a single sigmoid output. Our base detector, fine-tuned on BDD100K, achieved an mAP of 0.52 (at IoU 0.5), which is consistent with prior baselines and confirms the solid foundation of our model.

### 4.2 ADVERSARIAL TRAINING SETUP

Adversarial examples were generated using the Fast Gradient Sign Method (FGSM) with perturbation strengths $\varepsilon \in [0.003, 0.02]$ randomly sampled from a uniform distribution. This method ensures the classifier learns to generalize across a wide range of attack magnitudes, rather than just a few discrete values.

Training followed the two-phase strategy: **Phase 1:** Only the adversarial classification head was trained for 50 epochs, with the detector frozen. A strong adversarial weight of $\lambda_{\text{aux}} = 2.0$ and a learning rate of $1 \times 10^{-4}$ were used. **Phase 2:** The entire network was fine-tuned jointly for 35 epochs, with $\lambda_{\text{aux}} = 1.0$ and a reduced learning rate of $1 \times 10^{-5}$. A ReduceLROnPlateau scheduler decreased the learning rate if validation stalled.

Training employed AdamW with a batch size of 24. Data loaders used 7 worker threads, persistent workers enabled, and shuffling during training. All models were implemented in PyTorch with PyTorch Lightning and trained on a Google Colab Pro T4 GPU (25GB RAM). For runtime evaluation, inference was also benchmarked on a MacBook Pro (13-inch, 2020) with a ResNet-50 backbone. On average, inference on clean images took ∼0.7s. The total time for generating an adversarial example and running subsequent inference ranged from ∼1.7s (FGSM) to ∼9.5s (BIM), confirming the lightweight overhead of our additional classification head.

Table 1: Relative performance degradation (%) of FasterRCNN under adversarial attack ($\epsilon = 0.003$) compared to clean conditions. All metrics were computed using a confidence threshold of 0.7 and an IoU threshold of 0.5.

| Attack | mAP@0.5 ↓ | mAP@.5:.95 ↓ | Prec. ↓ | Rec. ↓ | F1 ↓ | IoU ↓ |
|--------|-----------|--------------|---------|--------|------|-------|
| FGSM   | 42.5%     | 56.6%        | 27.5%   | 24.8%  | 26.2% | 7.2% |
| BIM    | 59.0%     | 69.4%        | 33.9%   | 35.3%  | 34.6% | 7.2% |
| PGD    | 47.0%     | 58.0%        | 27.4%   | 29.1%  | 28.3% | 6.0% |

## 5 RESULTS

This section presents a comprehensive evaluation of our proposed two-stage training framework, focusing on two primary objectives: (1) to improve object detection performance under adversarial noise and (2) to identify adversarial inputs using the dedicated binary classifier reliably. We detail quantitative metrics, ROC analysis, and qualitative visualizations to demonstrate the effectiveness and robustness of our approach.

### 5.1 DETECTION PERFORMANCE ON CLEAN AND PERTURBED INPUTS

Figure 3 illustrates the effect of increasing adversarial strength ($\epsilon$) on object detection performance in both training phases. All metrics—*mAP@0.5*, *mAP@[.5:.95]*, precision, recall, F1 score, and average IoU—decline with stronger perturbations, highlighting the increasing challenge of accurate detection under attack.

Phase 1 shows a sharp performance drop at low $\epsilon$, whereas Phase 2 is more robust. Metrics are computed on 20 validation images for the figure and 200 images for Table 1.

At $\epsilon = 0.003$, all attacks reduce detection performance. BIM causes the largest drop in *mAP@0.5* (59.0%) and F1 (34.6%), while PGD and FGSM also degrade performance. Average IoU drops consistently by 6–7% (Table 1).

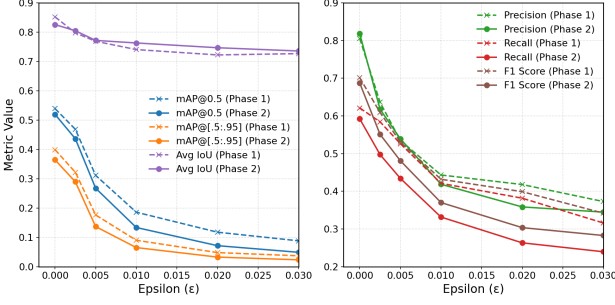

Figure 3: Object detection performance under adversarial perturbations ($\epsilon$), comparing Phase 1 (dashed lines with *x* markers) and Phase 2 (solid lines with circle markers). Metrics are grouped into three subplots: mAP metrics, precision & recall, and average IoU & F1 score.

### 5.2 ADVERSARIAL CLASSIFICATION PERFORMANCE

We evaluated the binary classifier across perturbation strengths ($\epsilon$) using accuracy, precision, recall, and F1 score.

Figure 4 summarizes detection performance across perturbation strengths. Panel (a) shows that F1 scores exceed 0.85 from $\epsilon = 0.003$ onward, peaking near $\epsilon = 0.006$ where precision and recall are well balanced. At very low perturbations ($\epsilon = 0.0005$–$0.001$), detection is more challenging, although optimized thresholds mitigate this drop. Panel (b) presents ROC curves for FGSM: the AUC increases from 0.75 at $\epsilon = 0.0005$ to 0.97 for $\epsilon \geq 0.007$, confirming effective detection at stronger perturbations.

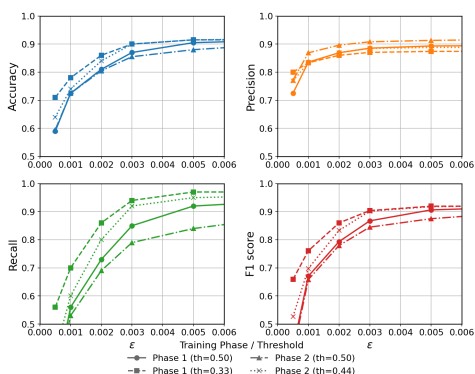 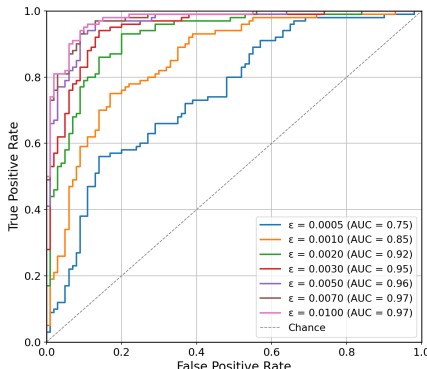

Figure 4: Detection performance across perturbation strengths. (a) F1 scores exceed 0.85 from $\epsilon = 0.003$, peaking near $\epsilon = 0.006$ where precision and recall balance. (b) ROC curves improve with higher perturbations, with AUC rising from 0.75 at $\epsilon = 0.0005$ to 0.97 for $\epsilon \geq 0.007$.

Table 2: Binary classification performance under clean and adversarial conditions.

| Phase | Attack | Threshold | AUROC | Accuracy | F1-score | Precision | Recall |
|-------|--------|-----------|-------|----------|----------|-----------|--------|
| P1 | FGSM | Default (0.50) | 0.944 | 87.0% | 0.867 | 0.885 | 0.850 |
| | | Best (0.44) | 0.944 | **90.0%** | **0.902** | 0.885 | **0.920** |
| | BIM | Default (0.50) | 0.986 | 93.0% | 0.933 | 0.898 | **0.970** |
| | | Best (0.63) | 0.986 | **95.5%** | **0.954** | **0.969** | 0.940 |
| | PGD | Default (0.50) | 0.976 | 91.5% | 0.917 | 0.895 | **0.940** |
| | | Best (0.65) | 0.976 | **93.5%** | **0.932** | **0.978** | 0.890 |
| P2 | FGSM | Default (0.50) | 0.947 | 85.5% | 0.845 | **0.908** | 0.790 |
| | | Best (0.33) | 0.947 | **90.0%** | **0.904** | 0.870 | **0.940** |
| | BIM | Default (0.50) | 0.987 | 93.5% | 0.936 | 0.922 | 0.950 |
| | | Best (0.53) | 0.987 | **94.5%** | **0.945** | **0.941** | 0.950 |
| | PGD | Default (0.50) | 0.979 | 92.0% | 0.920 | **0.920** | 0.920 |
| | | Best (0.43) | 0.979 | **93.0%** | **0.931** | 0.913 | **0.950** |

Table 2 summarizes binary classification performance under FGSM, PGD, and BIM attacks at $\epsilon = 0.003$ (PGD and BIM used 5 iterations with $\alpha = 0.0006$) for both Phase 1 and Phase 2. Even at this modest perturbation, F1 scores exceed 0.85 across all settings. Optimized thresholds further improve recall while maintaining a strong balance with precision.

AUROC remains high (0.944–0.987) across all attacks and phases, confirming strong discrimination. FGSM, as a single-step method, exhibits slightly lower performance than PGD and BIM but remains competitive after threshold tuning. Phase 1 generally requires higher optimal thresholds than Phase 2, likely due to its frozen backbone encouraging sharper decision boundaries. Phase 2, with joint backbone adjustment, achieves better calibration and effective detection with lower thresholds.

## 5.3 CROSS-DATASET GENERALIZATION

To assess robustness and generalization, we evaluated the Phase 2 BDD100K model on two additional autonomous driving datasets: KITTI and nuScenes. These datasets differ in acquisition hardware, environmental context, and temporal dynamics.

As shown in Table 3, the model achieves strong zero-shot transfer, with AUROC up to 0.865 on KITTI and 0.974 on nuScenes. Although performance is lower under FGSM, detection remains reliable across both datasets. Fine-tuning on KITTI and nuScenes further boosts performance, with

Table 3: Cross-dataset adversarial detection performance on KITTI and nuScenes. Results are reported for zero-shot transfer (model trained on BDD100K, Phase 2) and after fine-tuning on the target dataset.

| Dataset | Setting | Attack | AUROC | Accuracy | F1-score | Precision | Recall |
|---------|---------|--------|-------|----------|----------|-----------|--------|
| KITTI | Zero-shot | FGSM | 0.774 | 68.0% | 0.673 | 0.688 | 0.660 |
| | | PGD | 0.825 | 72.0% | 0.725 | 0.712 | 0.740 |
| | | BIM | 0.865 | 73.5% | 0.744 | 0.720 | 0.770 |
| | Fine-tuned | FGSM | 0.935 | 80.0% | 0.799 | 0.806 | 0.800 |
| | | PGD | 0.962 | 85.5% | 0.855 | 0.856 | 0.855 |
| | | BIM | 0.984 | 90.5% | 0.905 | 0.907 | 0.905 |
| nuScenes | Zero-shot | FGSM | 0.894 | 80.5% | 0.780 | 0.896 | 0.690 |
| | | PGD | 0.956 | 88.0% | 0.875 | 0.913 | 0.840 |
| | | BIM | 0.974 | 90.5% | 0.904 | 0.918 | 0.890 |
| | Fine-tuned | FGSM | 0.942 | 83.5% | 0.834 | 0.851 | 0.835 |
| | | PGD | 0.982 | 91.0% | 0.910 | 0.912 | 0.910 |
| | | BIM | 0.990 | 96.0% | 0.960 | 0.961 | 0.960 |

AUROC exceeding 0.93 under FGSM and surpassing 0.98 under PGD and BIM. F1-scores, precision, and recall also remain consistently high, highlighting the dual strength of our approach: strong cross-dataset transferability and efficient domain adaptation.

## 5.4 ABLATION STUDIES

We conducted extensive ablation experiments to investigate the impact of various training strategies and model design choices on the robustness of object detection and the performance of adversarial classification. The main findings are summarized as follows:

- **Training Strategy:** Employing shorter, phased training epochs in a fixed data subset consistently outperformed longer training schedules in larger and more varied subsets, indicating the benefits of focused learning phases.

- **Joint vs. Separate Training:** Training the adversarial classifier separately on frozen detector features led to faster convergence and improved early-stage performance. In contrast, simultaneous joint training with both detection and adversarial losses from the outset resulted in slower convergence, likely due to negative task interference.

- **Two-Phase Learning Scheme:** Initially freezing the backbone during adversarial classifier training, then unfreezing it for fine-tuning in a subsequent fine-tuning phase, enhanced generalization capabilities while preserving training stability.

- **Threshold Optimization:** Careful adjustment of the classification threshold significantly improved detection rates for low-strength adversarial examples, demonstrating the importance of threshold calibration.

## 5.5 QUALITATIVE RESULTS

To further examine model behavior, we visualize samples from clean and adversarial conditions, categorized into three types:

- **Clean (Correct)**: Serve as baseline references; bounding boxes and scene labels align well with ground truth.

- **Adversarial (Correct)**: Scene classification remains correct, but detection quality often degrades: bounding boxes shift or scale abnormally.

- **Adversarial (Incorrect)**: Classification fails, and object detection is further compromised by distorted or missing bounding boxes.

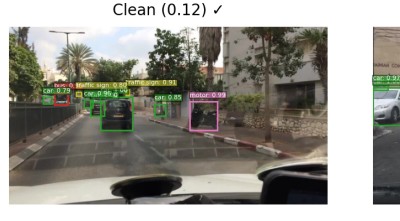 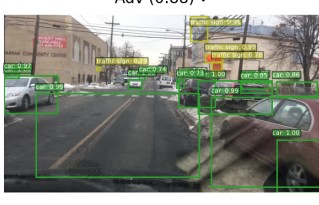 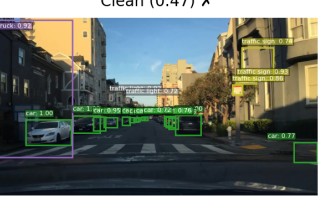

(a) Clean sample      (b) Correct adv. detection      (c) Incorrect adv. detection

Figure 5: Qualitative examples of clean (a), correctly detected adversarial (b), and misclassified adversarial (c) samples from the BDD100K dataset. (a) Clean sample: object detector and adversarial classifier both correct (score 0.12). (b) Correct adversarial detection (FGSM, score 0.88): object detection degrades, with imprecise or spurious boxes. (c) Misclassified adversarial case (FGSM, score 0.47): adversarial classifier fails while perturbations cause multiple false positives.

Examples of these cases are shown in Figures 5a–5c, demonstrating clean, correctly classified adversarial, and misclassified adversarial samples.

Figure 6 presents a comparison of the output of the object detection model on adversarial inputs generated by FGSM, PGD, and BIM attacks, all in the same perturbation budget ($\epsilon = 0.003$). Consistent with our observations, all attacks exhibit similar disruptive effects: bounding box centers tend to drift, object sizes are distorted (e.g., some car boxes become excessively large), and class predictions shift. For instance, cars originally labeled as "car" (green) are often misclassified as "truck" (violet), "bus" (red), or "motorcycle" (pink). Moreover, adversarial perturbations lead the model to hallucinate trucks in building structures or identify traffic signs within advertisement banners. These examples vividly highlight how even subtle perturbations can significantly degrade both classification and localization performance.

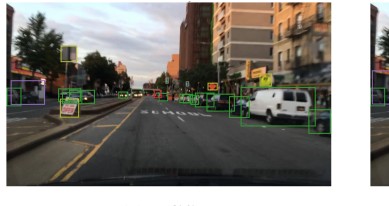 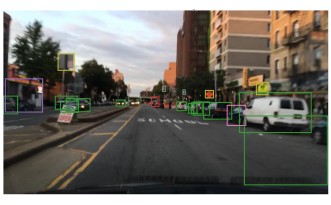 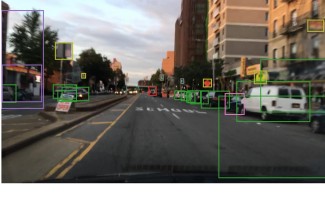

(a) FGSM      (b) PGD      (c) BIM

Figure 6: Adversarial examples generated by FGSM ($\epsilon = 0.003$), PGD ($\epsilon = 0.003$, iters=5, $\alpha = 0.0006$), and BIM ($\epsilon = 0.003$, iters=5, $\alpha = 0.0006$) at comparable perturbation levels.

We also observe distinct distributions of classification confidence:

- **Clean (correct):** $\mu = 0.172$, $\sigma = 0.125$
- **Adversarial (correct):** $\mu = 0.776$, $\sigma = 0.123$
- **Adversarial (misclassified):** $\mu = 0.326$, $\sigma = 0.124$

These distributions suggest that adversarial examples are harder to classify when their confidence scores fall in ambiguous regions.

To quantify this degradation, we measured the relative change in the area of the boundary box and the shift in the center between clean and adversarial predictions across multiple strengths of perturbation ($\epsilon = 0.001, 0.003, 0.01$). The metrics are defined as follows:

$$\text{Area Change} = \frac{|\mathcal{A}_{\text{adv}} - \mathcal{A}_{\text{clean}}|}{\mathcal{A}_{\text{clean}}} \tag{2}$$

$$\text{Center Shift} = \sqrt{(x_{\text{adv}} - x_{\text{clean}})^2 + (y_{\text{adv}} - y_{\text{clean}})^2} \tag{3}$$

where $\mathcal{A}$ denotes the boundary box area and $(x, y)$ are the center coordinates. We compute both metrics per instance and report class-wise averages.

As shown in Figure 7, both metrics increase consistently with higher $\epsilon$. For example, at $\epsilon = 0.01$, the boundary box area for certain classes (e.g., *bus*, *motor*) changed by over 30%, and center shifts became more pronounced. These results demonstrate that even when classifications remain correct, the localization accuracy of object detection deteriorates with adversarial noise, reinforcing that perturbations disrupt spatial features differently across tasks.

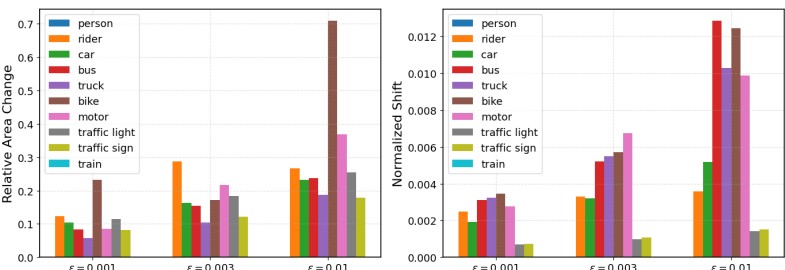

Figure 7: Impact of adversarial perturbations on bounding box outputs. Left: Relative change in box area. Right: Normalized center shift. Values are aggregated over 10 classes.

## 6 DISCUSSION

Our results show that even small imperceptible perturbations ($\epsilon = 0.003$) significantly degrade object detection, causing reduced confidence, bounding box drift, and missed or spurious detections, highlighting vision systems' vulnerability.

The two-phase training strategy, with a frozen detector followed by joint fine-tuning, stabilizes learning and improves the robustness and calibration of the adversarial classifier. Notably, the classifier generalizes well: trained only on FGSM perturbations, it performs effectively against stronger iterative attacks, suggesting it learns generalizable adversarial features rather than attack-specific patterns.

Unlike existing defenses, our approach integrates a modular adversarial head directly into the detector backbone, offering a lightweight, attack-agnostic solution compatible with large-scale pipelines. Cross-dataset experiments on KITTI and nuScenes confirm its transferability and practical applicability.

The observed decoupling between detection and adversarial classification metrics underscores the benefit of explicit supervision: dedicated adversarial signals complement detection outputs and point toward hybrid defense strategies that maintain clean performance while improving robustness.

**Limitations.** Our method has been evaluated only on Faster R-CNN and white-box digital attacks; extending it to other detectors and adaptive or physical attacks is future work. While cross-dataset experiments show adaptability, we used reduced training subsets; scaling to full datasets and longer schedules remains to be explored.

## 7 CONCLUSION

We introduced a lightweight, two-stage framework to enhance adversarial robustness in automotive object detectors by integrating a binary classifier into the Faster R-CNN backbone. The classifier, trained in two distinct phases, initial training with a frozen detector followed by joint fine-tuning, achieved over 90% accuracy in detecting adversarial examples.

Crucially, our defense demonstrates strong zero-shot transferability across datasets (BDD100K, KITTI, nuScenes) and generalizes effectively to a range of attacks, including FGSM, BIM, and PGD. After fine-tuning, the model achieved an AUROC greater than 0.98, with minimal impact on object detection accuracy. The modular and efficient nature of our approach makes it highly suitable for real-world deployment in autonomous systems.

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
