# OpenReview forum: "Automotive Object Detection under Adversarial Attacks: A Robust Two-Stage Training Framework"
_ICLR.cc/2026/Conference — Submitted to ICLR 2026_

### Official Review · Reviewer_rL22 · 2025-10-30

**Soundness:** 2
**Presentation:** 2
**Contribution:** 1
**Rating:** 2
**Confidence:** 5

**Summary:**

The paper proposes a lightweight, modular two-stage training framework for detecting adversarial perturbations in automotive object detection. A binary adversarial classifier is attached to intermediate features of a Faster R-CNN backbone. Training proceeds in two phases: first, freezing the detector and training the classifier; second, joint fine-tuning with a weighted auxiliary loss. The method achieves high adversarial example detection accuracy on BDD100K and demonstrates cross-dataset transfer to KITTI and nuScenes, with further gains achieved after fine-tuning. The defense claims minimal impact on clean detection accuracy and low overhead.

**Strengths:**

1. Comprehensive Experimental Evaluation: The paper provides a thorough empirical analysis across multiple autonomous driving datasets (BDD100K, KITTI, nuScenes) and includes both quantitative metrics and qualitative visualizations.

2. Practical Two-Phase Training Strategy: The staged training approach (frozen detector followed by joint fine-tuning) is well-motivated and shows improved stability compared to simultaneous training, as demonstrated in the ablation studies.

3. Detailed Quantitative Analysis: The paper includes extensive metrics beyond simple accuracy, including bounding box drift analysis, area change measurements, and confidence distribution analysis, which provide insights into how adversarial perturbations affect detection beyond classification errors.

4. Modular and Lightweight Design: The proposed defense maintains compatibility with pre-trained models and adds minimal computational overhead, which is important for practical deployment.

**Weaknesses:**

1. Misleading "Attack-Agnostic" Claim: While the authors claim their method is "attack-agnostic," it is only trained on FGSM and evaluated on a narrow set of gradient-based white-box attacks (FGSM, BIM, PGD). The generalization to truly unseen attack types (e.g., C&W, spatial transformations, patch attacks, frequency-domain perturbations) is not demonstrated. The claim that training on FGSM alone provides generalization to "attack-agnostic" defense is overstated: the tested attacks share similar gradient-based mechanisms. More general defenses based on spatial-temporal consistency or input transformations would be more deserving of this label.

2. Limited Architectural Scope: The method is developed and evaluated exclusively on Faster R-CNN, a relatively outdated 2D detector. Modern autonomous driving systems increasingly rely on 3D object detection frameworks (e.g., PointPillars, CenterPoint, BEVFusion), Multimodal fusion approaches (e.g., LiDAR + camera), and End-to-End planning models. The lack of evaluation on modern architectures significantly limits the practical relevance and generalizability of this work.

3. Insufficient Novelty: The core contribution, attaching a binary classifier for adversarial detection, is highly incremental. Similar detection-based defenses have been extensively studied in image classification (Metzen et al., 2017; Feinman et al., 2017). The extension to object detection does not introduce substantial technical innovation. The two-phase training is a reasonable engineering choice but not a fundamental contribution.

4. Missing Baseline Comparisons: The paper fails to compare against established adversarial detection methods, including Feature Squeezing, MagNet, Local Intrinsic Dimensionality, Consistency-based defenses, and Ensemble detection methods.

5. Weak Threat Model: The assumption that adversaries have white-box access to the detector but not the adversarial classifier is unrealistic. In practice, adaptive attackers would target the entire system, the classifier's architecture and parameters could be reverse-engineered, and detection-based defenses are known to be vulnerable to adaptive attacks. The paper provides no evaluation against adaptive attacks that specifically target the detection mechanism.

6. Limited Dataset Scale: While the authors claim to use BDD100K, KITTI, and nuScenes, they actually use only ~2% of BDD100K (~1,400 images) and similarly small subsets of other datasets (< 1,500 images each). This severely limits: statistical significance of the results, confidence in generalization claims, and practical relevance for real-world deployment. The justification of "reduced computational cost" is insufficient: robust adversarial defense evaluation requires proper scale.

7. Questionable Design Choices: Why use global average pooling on backbone features rather than RPN or RoI features, which contain more task-relevant information? The fixed threshold calibration is fragile and deployment-specific. No discussion of false positive rates in safety-critical scenarios. Evaluation Metrics: The paper focuses heavily on AUROC and F1-score for adversarial detection, but provides limited analysis of: false positive rates at safety-critical operating points, trade-offs between clean performance and robust performance
computational overhead in realistic deployment scenarios, and the latency impact on real-time processing requirements.

**Questions:**

Please see Weaknesses.

---

### Official Review · Reviewer_H6fC · 2025-10-31

**Soundness:** 2
**Presentation:** 2
**Contribution:** 1
**Rating:** 2
**Confidence:** 5

**Summary:**

The paper proposes a lightweight two-stage framework for adversarial example detection in automotive object detection. The core idea is to attach a binary adversarial classifier to intermediate features of a Faster R-CNN backbone, first training the detector-frozen classifier on clean/adversarial samples (Phase 1), then jointly fine-tuning the whole network (Phase 2). Experiments on BDD100K, with transfer to KITTI and nuScenes, show high AUROC against digital white-box attacks (FGSM, BIM, PGD) while claiming minimal degradation of detection accuracy.

**Strengths:**

1. Problem is meaningful: adversarial robustness for perception in autonomous driving is still an open and safety-critical setting.

2. The proposed defense is modular and detector-attached, so in principle it can be plugged into existing detection pipelines without redesigning the detector.

3. Two-phase training is reasonable to stabilize multi-task optimization and is empirically validated.

**Weaknesses:**

1. Attack space is outdated and narrow. The paper only considers classical, early attacks (FGSM, BIM, PGD). For ICLR 2026, this is too weak: stronger and standardized suites such as AutoAttack / APGD, Square Attack, DI/MI/TI-FGSM, EOT-based attacks, and patch/expectation-based attacks for detectors should be evaluated.
﻿
2. Only one detector (Faster R-CNN) is used. Current automotive perception stacks and research benchmarks routinely use anchor-free/one-stage and transformer-based detectors (YOLO, RT-DETR, DINO/DETR, et al.).
﻿
3. There are many detector-style defenses for adversarial examples that the paper does not position against, e.g. feature squeezing (Xu et al. 2017), MagNet, NIC (Ma et al.), Mahalanobis-based detectors, LID (Ma et al.), ML-LOO / outlier exposure–style detectors, even more recent self-supervised–feature detectors. The proposed method is architecturally very close to “attach an auxiliary binary head to deep features,” which is a well-trodden line; the paper needs to clarify what is new beyond applying it to Faster R-CNN.
﻿
4. Scenario-method mismatch. The paper anchors the motivation on autonomous driving, but then explicitly ignores the two most practical threat models in this domain: (i) physical attacks (adversarial patches on signs/vehicles, printable patterns, ShapeShifter, AdvPatch for detectors) and (ii) black-box attacks (which are realistic in V2X/cloud pipelines). For automotive, it is hard to justify evaluating only digital, white-box attacks.
﻿
5. “Why automotive?” remains unanswered. The described module (attach a binary head to intermediate features; two-stage training) is general and can be plugged into non-automotive detection (e.g. COCO, aerial, multi-modal). If the selling point is “automotive,” the paper should bring automotive-specific attacks (physical, sensor-level corruptions, black-box, patch-on-lane/sign) or automotive-specific constraints (latency on ECU, robustness across weather/time-of-day).

**Questions:**

1. If the method is “modular,” can it be attached to YOLO/RT-DETR/DINO without retraining the whole detector?

2. If the authors insist on the automotive scope, why are physical-world perturbations out of scope?

3. In an automotive setting, attackers are often black-box or partially black-box. What happens if the attacker does not know the auxiliary head and only attacks the detector, or performs transfer from a surrogate?

---

### Official Review · Reviewer_EQJr · 2025-10-31

**Soundness:** 2
**Presentation:** 3
**Contribution:** 2
**Rating:** 2
**Confidence:** 4

**Summary:**

This paper proposes a two-stage training framework to enhance adversarial robustness in automotive object detection. The method integrates a lightweight binary classifier into the Faster R-CNN backbone to detect adversarial inputs

**Strengths:**

Well-motivated two-stage training strategy: The phased approach (frozen detector → joint fine-tuning) is intuitive and addresses the challenge of balancing adversarial robustness with clean performance. The ablation study confirms this design choice is effective.

Comprehensive cross-dataset validation: Testing on three diverse autonomous driving datasets (BDD100K, KITTI, nuScenes) demonstrates both zero-shot transferability and adaptation capabilities, which is valuable for practical deployment.

Thorough experimental analysis: The paper includes multiple evaluation dimensions—ROC curves, threshold optimization, bounding box drift analysis, and qualitative visualizations—providing good insight into model behavior.

Lightweight and modular design: The auxiliary classifier adds minimal computational overhead (≈0.7s clean inference vs. 1.7-9.5s with attack generation), making the approach practically viable for real-time systems.

Strong detection performance on evaluated attacks: Achieving AUROC > 0.98 after fine-tuning on BIM/PGD attacks with F1-scores exceeding 0.90 demonstrates effective adversarial detection capability within the tested scenarios.

**Weaknesses:**

1. Severely limited experimental scale: Using only 2% of the training data (≈1,400 images) across all datasets is a critical flaw that undermines the validity of all conclusions.
2. Unrealistic threat model and missing adaptive attacks: The assumption that attackers can access the detector but not the adversarial classifier is implausible in real white-box scenarios.
3. No comparison with existing defense methods: The paper evaluates only its own approach without comparing against established defenses 4. like adversarial training, input transformations, or other detection-based methods.
4. Limited architectural scope: Evaluation only on Faster R-CNN limits generalizability claims.
5.  While the paper claims "minimal impact on object detection accuracy," Table 1 shows 42-59% mAP degradation under attack.

**Questions:**

1. Can you provide results using significantly more training data? The current use of only 2% of each dataset severely limits confidence in the findings. What are the performance trends when scaling to 25%, 50%, or ideally 100% of the training data? Are the reported generalization capabilities maintained at scale?
2. Your threat model assumes attackers cannot access the adversarial classifier, but in realistic white-box scenarios this is implausible. Can you evaluate against adaptive attacks such as BPDA (Backward Pass Differentiable Approximation) or attacks that jointly optimize against both the detector and classifier?
3. Without baselines, it's unclear if your approach offers advantages. Can you compare against adversarial training (e.g., Madry et al. 2017 applied to detection), input preprocessing defenses, or other detection-based methods under the same experimental setup?
4. Testing only on Faster R-CNN is limiting. Can you demonstrate that the two-stage training approach works on at least one other contemporary detector (e.g., YOLOv8, DETR, or newer architectures) to validate architectural independence?
5.  Please clarify: (a) Does the adversarial classifier cause any mAP degradation on clean validation images? (b) What is the false positive rate where clean images are incorrectly flagged as adversarial? (c) How do you recommend handling images flagged as adversarial in deployment scenarios?

---

### Official Review · Reviewer_FhEs · 2025-10-31

**Soundness:** 3
**Presentation:** 3
**Contribution:** 1
**Rating:** 2
**Confidence:** 5

**Summary:**

The paper presents a lightweight, two-stage framework to enhance adversarial robustness in automotive object detectors by integrating a binary classifier into the Faster R-CNN backbone. The classifier was trained in two distinct phases, initial training with a frozen detector followed by joint fine-tuning. It achieved over very high accuracy in detecting adversarial examples.

**Strengths:**

1. The paper is clearly written and easy to understand.
2. To the best of my knowledge, it is the first work that tests adversarial robustness of an object detector on autonomous datasets BDD100K, KITTI and nuScenes.
3. The detection results of adversarial examples are very good. And its generalization ability across datasets is high.

**Weaknesses:**

1. The proposed method looks simple and naive, just an integration of a binary classifier for adversarial examples and benign examples into an object detector. Both the classifier and the detector are very traditional methods.
2. It is not surprising that the classification accuracy was high, as this was shown in numerous papers many years ago. Moreover, the setting is  favorable for defense because it is assumed that the adversary does not have access to the proposed detection module.
3. The adversarial examples were generated in the digital world, which makes its practical value questionable. For an autonomous vehicle, a more realistic threat is the adversarial example in the physical world.

**Questions:**

L236-237: It is said that Fig. 3 presents results in both training phases. Are they training results or testing results?
L240: it is said that Phase 2 is more robust. But from Fig. 3, Phase 2 results are in general lower than Phase 1 results. Please explain.

---

### Meta-Review · Area_Chair_dD7r · 2026-01-06

**Summary:**

The reviewers are in strong agreement that, despite clear presentation and solid empirical execution within a constrained setting, the submission does not meet the bar for acceptance. The paper addresses adversarial robustness in automotive object detection by integrating a lightweight binary adversarial classifier into a Faster R-CNN detector using a two-stage training strategy. Across reviews, the contribution is viewed as highly incremental, with the core idea closely aligned with well-established adversarial detection paradigms, and the empirical claims are weakened by several aspects: a narrow and outdated attack model, unrealistic threat assumptions, limited dataset scale, and evaluation restricted to a single detector architecture. These concerns are consistently raised with high reviewer confidence, and all reviewers assign low ratings indicating rejection. The recommended decision is reject.

**Reviewer Concerns:**

There is not rebuttal.

**Reviewer Scores:**

All review scores are consistently low and with high confidence, so I strongly believe they will not change scores if they participate fully in the discussion.

---

### Decision · Program_Chairs · 2026-01-26

Reject